# The effect of whole-body vibration on lower extremity function in children with cerebral palsy: A meta-analysis

Xiaoye Cai[1], Guoping Qian[2], Siyuan Cai[3], Feng Wang[4], Yingjuan Da[1], Zbigniew Ossowski[2] *

1 Shanghai Normal University Tianhua College, Shanghai, P. R. China, 2 Gdansk University of Physical Education and Sport, Gdańsk, Poland, 3 Victoria university of Wellington, Wellington, New Zealand, 4 Shanghai Ocean University, Shanghai, P. R. China

* zbigniew.ossowski@awf.gda.pl

**Data Availability Statement:** All relevant data are within the paper and its Supporting Information files.

## Abstract

### Objective

The aim of this meta-analysis was to evaluate the effect of whole-body vibration training on lower limb motor function in children with cerebral palsy in randomized-controlled trials (RCTs).

### Methods

Two independent reviewers systematically searched the records of nine databases (PubMed, Cochrane, Web of Science, EMBASE, CNKI, etc.) from inception to December 2022. Tools from the Cochrane Collaboration were used to assess risk of bias. Standard meta-analyses were performed using Stata 16.0 and Revman 5.3. For continuous variables, the arms difference was calculated as the weighted mean difference (WMD) between the values before and after the intervention and its 95% confidence interval (95% CI).

### Results

Of the 472 studies identified, 13 (total sample size 451 participants) met the inclusion criteria. Meta-analysis showed that WBV training could effectively improve GMFM88-D [WMD = 2.46, 95% CI (1.26, 3.67), P<0.01] and GMFM88-E [WMD = 3.44, 95% CI (1.21, 5.68), P = 0.003], TUG [WMD = -3.17, 95% CI (-5.11, -1.24), P = 0.001], BBS [WMD = 4.00,95% CI (3.29, 4.71), P<0. 01] and the range of motion of ankle joint and the angle of ankle joint during muscle reaction in children with cerebral palsy. The effect of WBV training on 6MWT walking speed [WMD = 47.64, 95% CI (-25.57, 120.85), p = 0.20] in children with cerebral palsy was not significantly improved.

### Conclusion

WBV training is more effective than other types of conventional physical therapy in improving the lower limb motor function of children with cerebral palsy. The results of this meta-analysis strengthen the evidence of previous individual studies, which can be applied to the

**Funding:** The authors received no specific funding for this work.

**Competing interests:** The authors have declared that no competing interests exist.

**Abbreviations:** CP, cerebral palsy; WBV, whole-body vibration; RCTs, randomized controlled trials; PRISMA, Preferred Reporting Items for Systematic Reviews and Meta-analysis; GMFM88-D/E, Gross motor function measurement-88 in Zone D/E; 6MWT, 6-min walk test; TUG, Timed Up and GO test; BBS, Berg Balance Scale; Ankle-ROM, The range of motion of ankle joint; Ankle-R1/R2, The angle of ankle (1/2) joint during muscle reaction; WMD, weighted mean difference; 95% CI, 95% confidence intervals; BoNT-A, Botulinum toxin type A; WBVAO, whole-body vibration combined with action observation.

clinical practice and decision-making of WBV training and rehabilitation in children with cerebral palsy.

## Introduction

Cerebral palsy (CP) is long-standing dyskinesia of motor posture caused by chronic brain injury to the developing fetus or infant [1], and it frequently results in limited motor abilities in children, of whom spastic cerebral palsy is the most common [2, 3]. More than 50% of children have lower limb movement disorders, such as lower limb muscle contracture, ankle stiffness and deformity, hip and knee flexion when standing and walking, sharp foot crossing, etc., which seriously affect the normal growth development and motor function of children [1–4].

Whole-body vibration (WBV) training method is a new non-traditional training method [5], through external intervention, which allows subjects to generate adaptive responses to vibration stimulation through a specially designed vibration platform [6, 7]. Currently, it is used as a clinical treatment to increase muscle strength, usually involving a series of static or dynamic movements in a standing posture on a vibrating pad. It was originally used by elite athletes to increase speed and strength [8]. In recent years, this method has been widely popularized and applied in many fields abroad, and has also achieved some positive therapeutic and training effects. WBV appears to be a promising adjunct to conventional treatment involving CP patients. According to the studies evaluated by these authors, WBV may help to improve walking ability, walking speed, overall mobility, muscle mass and force production, and reduce spasticity [8]. Furthermore, some authors concluded that horizontal WBV training should be included in rehabilitation programs for children with CP, as it can improve their physical performance without harmful effects [9].

Studies have also confirmed that WBV training can improve the lower limb function of children with cerebral palsy [10, 11]. However, due to the various experimental design protocols of relevant studies regarding the application of WBV training in lower limb rehabilitation of cerebral palsy patients at this stage, the small number of study subjects and the uneven quality of the comprehensive review literature often lead to conflicting final study conclusions. Therefore, it is necessary to comprehensively and systematically evaluate the effect of WBV training on the lower limb rehabilitation of children with cerebral palsy.

This study intends to select randomized controlled trials on the impact of WBV training on lower limb motor function of children with cerebral palsy and evaluate the results of the consistent study by meta-analysis. To comprehensively and quantitatively evaluate the improvement effect of WBV training on lower limb function indexes of children with cerebral palsy, and to provide more reliable basis for clinical practice and decision-making of WBV training and rehabilitation of children with cerebral palsy.

## Methods

### Literature search

The research team conducted the meta-analysis in compliance with the guidelines of the Preferred Reporting Items for Systematic Reviews and Meta-analysis (PRISMA) [12]. Two researchers independently performed a comprehensive literature review. The reviewed databases included Web of Science, PubMed, Embase, Scopus, Cochrane, EBSCO, CNKI, Wanfang Database and VIP, which were systematically searched to identify relevant articles up to December 15, 2022. Additionally, references were incorporated retrospectively into the

literature to supplement access to the relevant literature. The combination of free words with theme words was performed in this study.

Search strategies were developed using Boolean logical operators, truncates, etc. for comprehensive searches. Search terms were used including: whole-body vibration, WBV, Cerebral Palsy, CP, Dystonic-Rigid Cerebral Palsy, Mixed Cerebral Palsy, Monoplegic Infantile Cerebral Palsy, Quadriplegic Infantile Cerebral Palsy, Rolandic Type Cerebral Palsy, Congenital Cerebral Palsy, Little Disease, Spastic Diplegia, Monoplegic Cerebral Palsy, Athetoid Cerebral Palsy, Dyskinetic Cerebral Palsy, Atonic Cerebral Palsy, Hypotonic Cerebral Palsy, Diplegic Infantile Cerebral Palsy, Spastic Cerebral Palsy, Child, Lower Limb, Membrum inferius, Lower Extremity.

Using PubMed as an example, see Table 1 for specific search strategies (see S1 Table for the rest of the search strategies).

## Eligibility criteria

**Types of participant.**  Children with clinical and laboratory diagnoses of cerebral palsy were selected as experimental subjects to study and evaluate the effect of whole-body vibration (WBV) training on their lower-limb motor function.

**Intervention measures.**  *Control group*. Conventional physiotherapy.

*Experimental group*, WBV training.

## Primary outcomes

(1) Gross motor function measurement-88 in Zone D (GMFM-88-D)

GMFM-88 is applicable for evaluating gross motor function in cerebral palsy, with appropriate reliability and validity. It encompasses the evaluation of five functional areas A-E, with each evaluated individually or in combination. A total of 13 items are found in Zone D, reflecting standing ability [13, 14].

(2) Gross motor function measurement-88 in Zone E (GMFM-88-E)

A total of 24 items exist in Zone E, reflecting the ability to walk, run and jump [14]. According to the completion level, each item scores 0–3 points [14]. A greater functional level brings about a higher score [15].

(3) Six-min walk test (6MWT)

The 6MWT was utilized to assess the walking speed of participants. The child patient stood on the starting line and set the timer going as soon as walking. Patients walked back and forth

**Table 1. Search strategy (PubMed).**

| Number | Search Strategy |
|---|---|
| #1 | ("whole body vibration"[MeSH Terms]) OR ("WBV"[All Fields]) |
| #2 | ("Cerebral Palsy"[MeSH Terms]) OR ("CP"[All Fields]) OR ("Dystonic-Rigid Cerebral Pals*"[All Fields]) OR ("Mixed Cerebral Pals*"[All Fields]) OR ("Monoplegic Infantile Cerebral Pals*"[All Fields]) OR ("Quadriplegic Infantile Cerebral Pals*"[All Fields]) OR ("Rolandic Type Cerebral Pals*"[All Fields]) OR ("Congenital Cerebral Pals*"[All Fields]) OR ("Little* Disease"[All Fields]) OR ("Little's Disease"[All Fields]) OR ("Spastic Diplegia"[All Fields]) OR ("Monoplegic Cerebral Pals*"[All Fields]) OR ("Athetoid Cerebral Pals*"[All Fields]) OR ("Dyskinetic Cerebral Pals*"[All Fields]) OR ("Atonic Cerebral Pals*"[All Fields]) OR ("Hypotonic Cerebral Pals*"[All Fields]) OR ("Diplegic Infantile Cerebral Pals*"[All Fields]) OR ("Spastic Cerebral Pals*"[All Fields]) |
| #3 | ("Children"[MeSH Terms]) OR ("child"[All Fields]) |
| #4 | ("Lower Limb"[MeSH Terms]) OR ("Membrum inferius"[All Fields]) OR ("Lower Extremit*"[All Fields]) OR ("Lower Limbs"[All Fields]) |
| #5 | #1 AND #2 AND #3 AND #4 |

as much as they could in the interval. During this period, the monitor could give verbal advice and encouragement. At the end of the test, the 6-min walking distances of patients were recorded [16, 17].

(4) Timed Up and Go test (TUG)

In the TUG test, participants wore ordinary shoes, sat on back chairs with armrests, and leaned on the back of chairs. They held hands on armrests, stood up from the back chair, and used walking aids if necessary. Child patients walked 3 meters forward with a usual walking gait, crossed the thick line or mark. Then they turned around and walked back to the chair. All patients sat down and leaned against the back of chairs. A stopwatch was used to record the time from the patient's back leaving the chair back to their sitting again. The risk of falling during the test was recorded. The test was repetitively conducted 3 times, with data recorded to calculate and take the average value for analysis [18, 19].

## Secondary outcomes

(1) Berg Balance Scale (BBS)

BBS was used to evaluate the balance function. Items totaled 14, with each scoring 0–4 points. A relatively high score represented greater balance ability [20, 21].

(2) The range of motion of the ankle joint (Ankle-ROM)

A joint protractor was mainly used to measure the range of motion (ROM) of the ankle joint. Also, ankle-ROM was divided into active and passive. Each measurement was performed 3 times, and the mean was recorded as ROM [22].

(3) The angle of ankle (1/2) joint during muscle reaction (Ankle-R1/R2)

Ankle-R1/R2-fast angle refers to the angle of jamming during rapid traction, recorded as angle 1 (R1); the slow angle refers to the full range of the joint motion under slow movement, recorded as angle 2 (R2) [11].

## Exclusion criteria

(1) History of bone and joint diseases of lower limbs, orthopedic surgery and use of antispasmodic drugs.

(2) Severe cognitive, intellectual and hearing impairment.

(3) Studies focused on therapy for lower-limb motor function in children with cerebral palsy rather than WBV training.

(4) Obvious errors in the original research and test scheme (incomplete article content, etc.).

(5) Non-randomized controlled trials (review literature, repetitive published clinical trial literature and animal trial research literature).

(6) Incomplete literature for data reporting.

(7) Studies with insufficient data failing to interpret results.

## Literature quality assessment and data extraction

**Data extraction and management.**  The literature review was screened separately by two scholars. Following the computer retrieval of major databases, titles and abstracts of all filtered articles were imported into the literature management software (EndNote X9 and Microsoft Excel). Duplicate literature was screened out. Titles and abstracts were checked to figure out articles unqualified for inclusion requirements. Finally, through full-text reading, literature meeting the criteria was sifted out. Any discrepancies during literature screening were solved by consulting relevant experts or talking to a third researcher. If numerous articles relevant to a study had been published, those with the most comprehensive experimental data and of

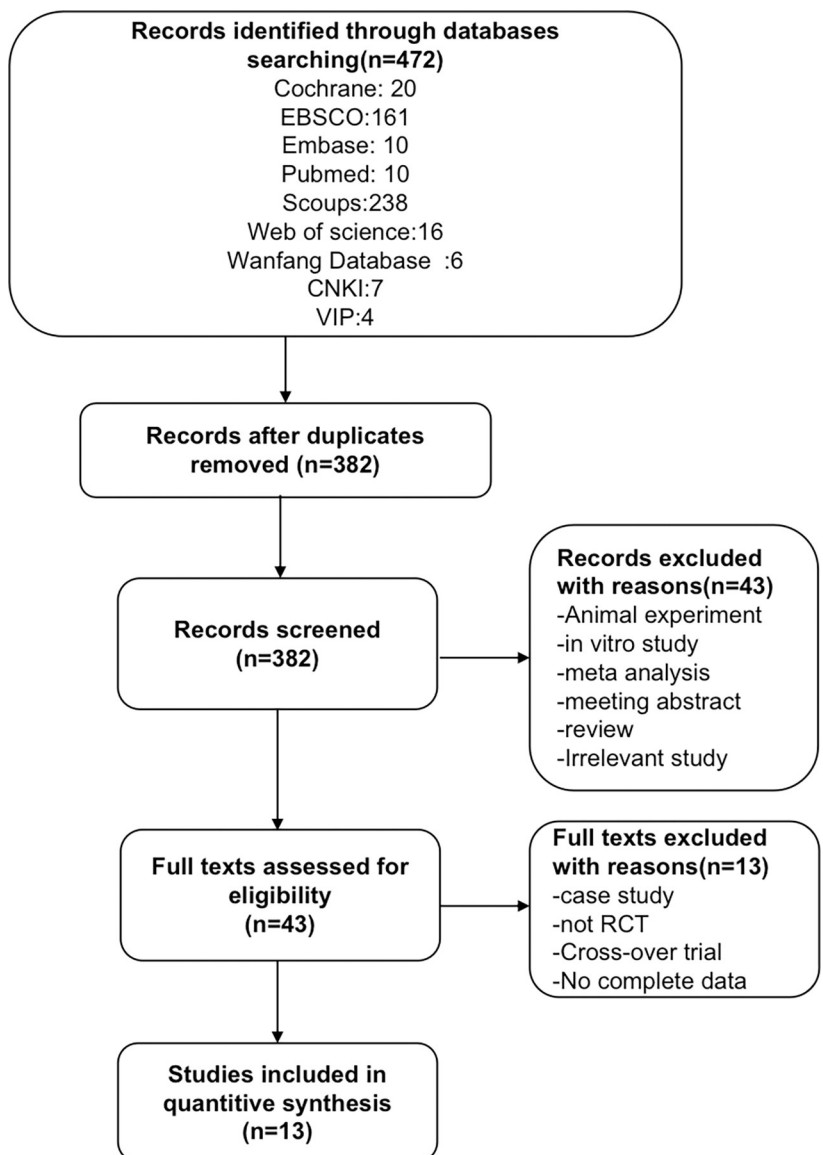

**Fig 1. The PRISMA flow chart of literature selection.**

outstanding consistency with inclusion criteria would be included in this study. Office program was used to sort out and analyze all the selected eligible literature in terms of basic literature information (title, author, and publication year), general information (patient age, gender, number of cases, etc.), experimental design (interventions), and outcome indicators. Literature with inconsistent outcome indicator units would be uniformly translated before data processing. When data were lacking in the literature, we would contact the original author by e-mail or phone to gather the necessary information. Fig 1 depicts the PRISMA flow chart for choosing literature.

**Literature quality assessment.** The findings of a bias risk assessment adopting the approach provided by the Cochrane Handbook for System Reviews of Interventions Version 5.1.0 were presented in the chosen literature. Evaluation can be conducted by resorting to the randomization method, allocation of covert scheme design, blind method, and reporting of

results data. Moreover, literature quality could also be assessed by verifying selective reporting of research results, other reasons of bias, etc. According to the results, "Yes" indicated the reasonable method or complete data and the low risk of bias; "Unclear" denoted unclear method and moderate risk of bias; "No" represented the incorrect method or incomplete data and high risks of bias. Finally, the examination findings were entered into the RevMan 5.3 program to generate a bias risk assessment chart.

## Statistical analysis

Meta-analysis was performed using RevMan 5.3 and STATA 16.0 (STATA Corp, College Station, TX, USA). For continuous variables, the weighted mean difference (WMD) was used as the effect index, and each effect size was expressed with a 95% confidence interval (95% CI), $P<0.05$ indicates a statistically significant difference between the two groups. The $I^2$ statistic was used to test the heterogeneity among different studies. When $P>0.1$ and $I^2 \leq 50\%$, it means that there was good homogeneity among all studies, and the fixed effect model was used to combine the effect size. When $P \leq 0.1$ and $I^2 > 50\%$, it means that there was heterogeneity between the studies, and the random effect model was used to combine the effect size. The source of heterogeneity was identified, and sensitivity analysis was conducted by article by article elimination. If $I^2 \leq 50\%$ after deleting a single study, it was considered that the study might be the source of influencing the combined effect size, and it was excluded from the meta-analysis. In addition, The Begg's and Egger's tests were performed to assess publication bias. $P<0.05$ was statistically significant unless otherwise specified.

## Results

### Search results

The S1 Table contains the complete search methods. Fig 1 depicts the study selection procedure. The nine databases yielded a total of 472 studies (PubMed = 10, Cochrane = 20, Scoups = 238, Web of Science = 16, Wanfang Database = 6, CNKI = 7, VIP = 4, Embase = 10, and EBSCO = 161). Other records could not be found through other sources. 382 articles remained after duplicates were deleted; however, 339 records were further deleted after reading the titles and abstracts. For different reasons, 43 full-text papers were read in depth for eligibility, and 30 articles were removed. Finally, this meta-analysis comprised 13 papers [10, 11, 23–33].

### General characteristics and quality evaluation of included studies

**Literature characteristics.** Of the 13 articles in this meta-analysis, 13 reported the use of WBV training, 12 reported the use of conventional physiotherapy. One reported that Botulinum toxin type A (BoNT-A) injection was combined with conventional physiotherapy. One reported the use of whole-body vibration combined with action observation (WBVAO) training combined with conventional physiotherapy, and one reported that conventional physiotherapy was changed to passive stretching exercise. The patients included in the study were children with cerebral palsy, they need WBV training to observe the effect of this treatment on the motor function of lower limbs. All control groups were also children with cerebral palsy. Except for one control group with WBV training, the rest of the latter group of patients only received conventional physiotherapy without WBV training. The characteristics of the included articles are reported in Table 2.

**Risk of bias of included literature.** The included studies were assessed for selection bias, performance bias, detection bias, attrition bias, and reporting bias. The word "random" was

**Table 2. Characteristics of included studies.**

| Study Year | Race | General Information [Number; Gender; Age (years old)] | GMFCS Level (Number) Experimental Group | GMFCS Level (Number) Control Group | experimental design Intervention Group | experimental design Control Group | Outcome Indicators |
|---|---|---|---|---|---|---|---|
| Yan 2021 [23] | Asian | Experimental group: 20; 9 male, 11 female; 4.55±1.23 Control group: 20; 8 male, 12 female; 4.80 ±1.00 | levelI: 2 levelII: 3 levelIII: 9 levelIV: 6 | levelI: 3 levelII: 5 levelIII: 7 levelIV: 5 | WBV training (12 weeks, 5 times/week, 15 min/session, 5~25Hz) + Conventional physiotherapy | Conventional physiotherapy (12 weeks, 5 times/week, 1 h/session) | GMFM88-D/E |
| Tekin 2021 [28] | Caucasian | Experimental group: 11; 6 male,5 female; 11.82±3.55 Control group: 11; 5 male, 6 female; 13.09 ±2.66 | | | WBV training (8 weeks, 3 times/week, 15 min/session, 15Hz, 3mm) + Conventional physiotherapy | Conventional physiotherapy (8 weeks, 2 times/week, 45 min/session) | GMFM88-D/E, TUG |
| Hegazy 2021 [33] | Caucasian | Experimental group: 20; 12 male, 8 female; 6.05 ± 1.5 Control group: 20; 10 male, 10 female; 5.85 ±1.46 | levelI: 7 levelII: 13 | levelI: 9 levelII: 11 | WBV training (8 weeks, 3 times/week, 10 min/session, 10~25Hz, 2mm) + Conventional physiotherapy | Conventional physiotherapy (8 weeks, 3 times/week, 1 h/session) | 6MWT |
| Jung 2020 [10] | Asian | Experimental group: 7; 3 male, 4 female; 9.00 ±3.26 Control group: 7; 3 male, 4 female;8.71 ±3.19 | levelI: 4 levelII: 2 levelIII: 1 | levelI: 3 levelII: 3 levelIII: 1 | WBVAO training (WBV, 4 weeks, 3 times/week, 30 min/session, 12~18Hz with action observation, 30 min/day) + Conventional physiotherapy | Conventional physiotherapy (4 weeks, 3 times/week, 30min/session) + WBV (4 weeks, 3 times/week, 30min/session) | GMFM88-D/E, 6MWT, TUG |
| Ren 2019 [24] | Asian | Experimental group: 30; 18 male, 12 female; 43.5±10.89 (months) Control group: 30; 16 male, 14 female; 43.0 ±10.65 (months) | levelI: 7 levelII: 16 levelIII: 7 | levelI: 9 levelII: 14 levelIII: 7 | WBV training (25 weeks, 5 times/week, 8~10 min/session, 26Hz) + Conventional physiotherapy combined with BoNT-A injection | Conventional physiotherapy +BoNT-A injection (25 weeks, 5 times/week, 20~40 min/session) | Ankle-R1/R2 |
| Yin 2019 [27] | Asian | Experimental group: 28; 17 male, 11 female; 47.1±8.7(months) Control group: 28; 19 mela, 9 female; 48.7 ±8.4(months) | levelI: 7 levelII: 21 | levelI: 6 levelII: 22 | WBV training (12 weeks, 5 times/week, 6~15 min/session, 12Hz, 4mm) + Conventional physiotherapy | Conventional physiotherapy (12 weeks, 5 times/week, 20~45 min/session) | GMFM88-D/E, BBS, Ankle-(active/passive) ROM |
| Lee 2019 [26] | Asian | Experimental group: 10; 5 male, 5 female; 7.3 ±1.89 Control group:10; 5 male, 5 female; 7.4±1.9 | levelI: 4 levelII: 6 | levelI: 4 levelII: 6 | WBV training (10 weeks, 5 times/week, 1 h/session, 12~24Hz) + Conventional physiotherapy | Conventional physiotherapy (10 weeks, 5 times/week, 1 h/session) | GMFM88-D/E |
| Ahmadizadeh 2019 [32] | Asian | Experimental group: 10; 6.9±2.46 Control group: 10; 8.1 ±1.93 | levelI: 2 levelII: 5 levelIII: 3 | levelI: 2 levelII: 5 levelIII: 3 | WBV training with passive static stretching exercises (6 weeks,3 times/week, 18 min/session,20~24Hz,2mm) | Passive static stretching exercises (6 weeks, 3 times/week, 18 min/session) | 6MWT, Ankle-(active/passive) ROM |
| Cai 2018 [25] | Asian | Experimental group: 15; 9 male, 6 female; 3.5~4.9 Control group: 15; 8 male, 7 female; 5.0~7.8 | levelII: 7 levelIII: 8 | levelII: 6 levelIII: 9 | WBV training (8 weeks,5 times/week, 9 min/session, 12~21Hz) + Conventional physiotherapy | Conventional physiotherapy (8 weeks, 5 times/week, 30 min/session) | BBS |

(*Continued*)

**Table 2.** (Continued)

| Study Year | Race | General Information [Number; Gender; Age (years old)] | GMFCS Level (Number) Experimental Group | GMFCS Level (Number) Control Group | experimental design Intervention Group | experimental design Control Group | Outcome Indicators |
|---|---|---|---|---|---|---|---|
| **Meng 2017** [11] | Asian | Experimental group: 38; 20 male, 18 female; 21.2±6.4(months) Control group: 37; 21 male, 16 female; 20.9 ±7.6(months) | levelI: 7 levelII: 17 levelIII: 14 | levelI: 9 levelII: 12 levelIII: 16 | WBV training (12 weeks, 5 times/ week, 8~10 min/session, 26Hz) + Conventional physiotherapy | Conventional physiotherapy (12 weeks, 5 times/ week, 20~40 min/ session) | BBS, Ankle-R1/ R2 |
| **Dudoniene 2017** [31] | Caucasian | Experimental group: 10; 6 male, 4 female; 8.56±1.07 Control group: 10; 5 male, 5 female; 8.70 ±0.90 | | | WBV training (3 weeks, 5 times/ week, 5~10 min/session, 15Hz) + Conventional physiotherapy | Conventional physiotherapy (3 weeks, 5 times/week, 45 min/session) | GMFM88-D/E |
| **Ko 2015** [30] | Asian | Experimental group: 12; 5 male, 7 female; 9.37±2.69 Control group:12; 5 male, 7 female; 9.52 ±2.16 | levelI: 5 levelII: 2 levelIII: 5 | levelI: 8 levelII: 4 levelIII: 0 | WBV training (3 weeks, 2 times/ week, 9 min/session, 20~24Hz) + Conventional physiotherapy | Conventional physiotherapy (3 weeks, 2 times/week, 30 min/session) | TUG |
| **Ibrahim 2014** [29] | Caucasian | Experimental group: 15; 9.63±1.41 Control group: 15; 9.63 ±1.41 | | | WBV training (12 weeks, 3 times/ week, 9 min/session, 12~18Hz, 4~6mm) + Conventional physiotherapy | Conventional physiotherapy (12 weeks, 3 times/ week, 1 h/session) | GMFM88-D/E, 6MWT, TUG |

mentioned in all studies, with only one article mentioning "coin tossing", one mentioning "simple randomization", and three mentioning "random number table method". Based on the method of randomization and concealment of allocation, all studies were determined to have a low risk of selection bias. Of the seven articles that described the evaluator blinding method, four had a low risk of detection bias and three had a high risk (as one person evaluated both groups). The evaluator blinding method of the remaining six studies was unclear. The integrity of results data in three articles was determined to have a high risk of attrition bias as subjects chose to withdraw from the experiment, however, the remaining articles had a low risk. The presence of reporting bias was determined as unclear in three articles, and low risk in the remaining articles. These results are visualized in Figs 2 and 3.

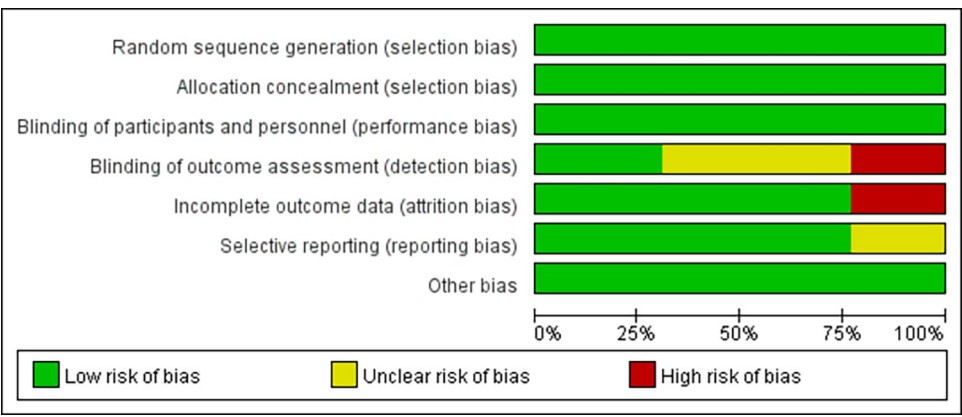

**Fig 2. Risk of bias graph.** The authors' judgments on each risk-of-bias item, presented as percentages across all included studies.

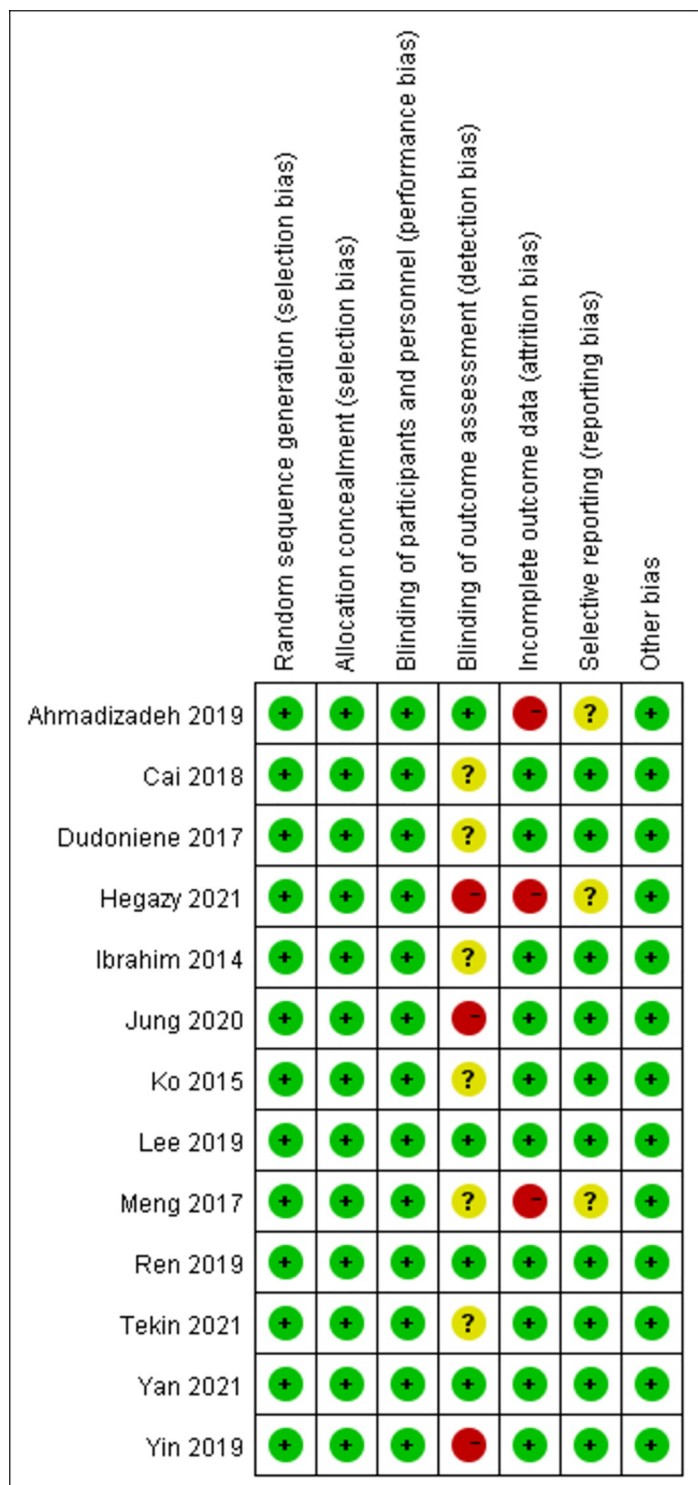

**Fig 3. Risk of bias summary.** The author's judgments on each risk of bias item for each included study.

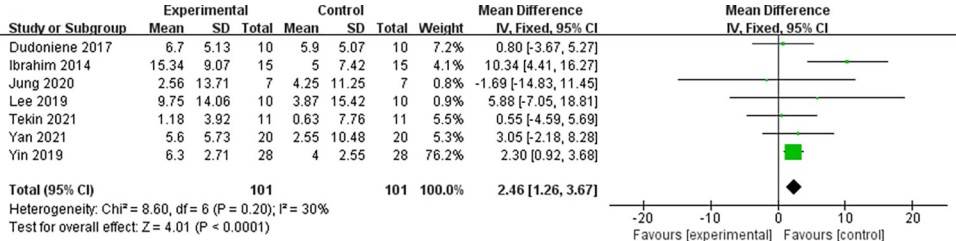

**Fig 4. Effect of WBV training on GMFM88-D score in children with cerebral palsy.**

## Meta-analysis results

**Effect of WBV training on GMFM88-D in children with cerebral palsy.** Seven of the included studies investigated the effect of WBV training on GMFM88-D indices in children with cerebral palsy. The results showed that, in comparison to the control group, WBV training could improve GMFM88-D in these children [WMD = 2.46, 95% CI (1.26, 3.67), Z = 4.01, P<0.01] (Fig 4). Egger's test showed no publication bias in these seven articles (P = 0.664>0.1). Further analysis showed that heterogeneity was not significant (P = 0.20, $I^2$ = 30%<50%). This leads us to conclude that WBV training combined with conventional physiotherapy could improve the lower limb gross motor function of children with cerebral palsy better than conventional physiotherapy alone.

**Effect of WBV training on GMFM88-E in children with cerebral palsy.** Seven of the included studies investigated the effect of WBV training on GMFM88-E indices in children with cerebral palsy. The results showed that, compared with the control group, WBV training could improve GMFM88-E these children [WMD = 3.44, 95% CI (1.21, 5.68), Z = 3.01, P = 0.003<0.01] (Fig 5). Egger's test showed no publication bias in these seven articles (P = 0.780>0.1) and heterogeneity was non-significant (P = 0.35, $I^2$ = 11%<50%). In summary, WBV training combined with conventional physiotherapy could improve the lower limb gross motor function of children with cerebral palsy better than conventional physiotherapy alone.

**Effect of WBV training on 6MWT in children with cerebral palsy.** Four studies investigated the effect of WBV training on 6MWT indices in children with cerebral palsy. Analysis of these studies revealed significant heterogeneity (P<0.01, $I^2$ = 89%). Sensitivity analysis was performed to determine the source of heterogeneity, however, we were unable to eliminate heterogeneity by excluding each article in turn. Thus, the random effect model was selected for analysis. The results showed that, compared with the control group, 6MWT scores were not improved by WBV training (Fig 6). Egger's test revealed no publication bias in the literature (P = 0.554>0.1).

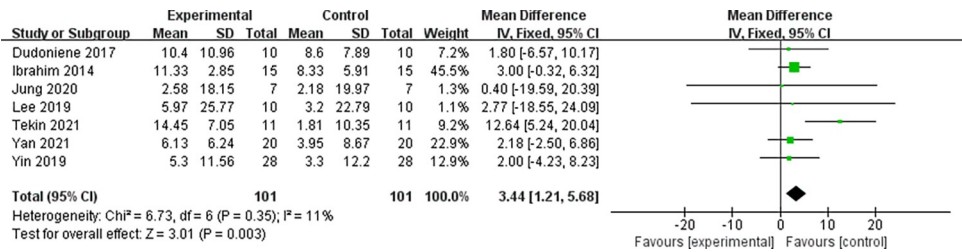

**Fig 5. Effect of WBV training on GMFM88-E score in children with cerebral palsy.**

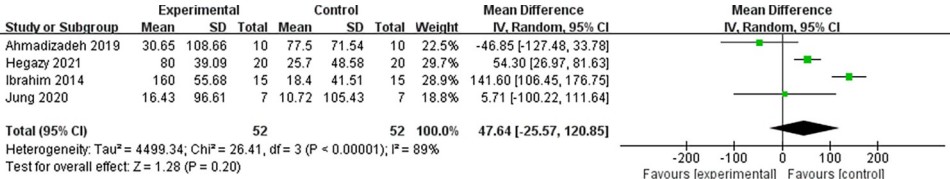

**Fig 6. Effect of WBV training on 6MWT walking speed of children with cerebral palsy.**

**Effect of WBV training on TUG in children with cerebral palsy.** Four studies investigated the effect of WBV training on TUG indices in children with cerebral palsy. The results showed that, compared with the control group, WBV training could reduce TUG time and risk of falls in these children [WMD = -3.17, 95% CI (-5.11, -1.24), Z = 3.22, P = 0.001] (Fig 7). Egger's test indicated no publication bias in these four articles (P = 0.436>0.1) and heterogeneity was not significant (P = 0.55, $I^2$ = 0.0%<50%). Thus, WBV training combined with conventional physiotherapy could improve TUG performance of children with cerebral palsy more effectively than conventional physiotherapy alone.

**Effect of WBV training on BBS in children with cerebral palsy.** For BBS, three studies investigated the effect of WBV training on BBS indices in children with cerebral palsy. Heterogeneity was significant (P = 0.00, $I^2$ = 96.0%). In order to further understand the source of heterogeneity, sensitivity analysis showed that the heterogeneity decreased significantly after excluding Cai's (2018) literature (P = 0.49, $I^2$ = 0.0%). The results showed that compared with the control group (S1 Fig), WBV training could significantly improve the balance ability of children with cerebral palsy [WMD = 4.00, 95% CI (3.29, 4.71), Z = 11.01, P<0.01]. In Begg's test, P = 0.317>0.1 indicated that there was no publication bias in the literature. In conclusion, WBV training combined with conventional physiotherapy was effective on improving the BBS balance ability of children with cerebral palsy than conventional physiotherapy alone.

**Effect of WBV training on Ankle function of children with cerebral palsy.** For ankle function, four studies investigated the effect of WBV training on ankle function indices in children with cerebral palsy. Two pieces of literature evaluated the active ROM of the ankle joint in children with cerebral palsy (P = 0.01, $I^2$ = 84%), and the result showed that there was a non-significant difference comparing with the control group (S2 Fig). The results of the passive ROM of the ankle joint in children with cerebral palsy [WMD = 3.78, 95% CI (0.74, 6.81), Z = 2.44, P = 0.01] showed that there were significant differences between the two groups (S3 Fig).

Two pieces of literature evaluated the ankle-R1 [WMD = 5.37, 95% CI (3.24, 7.50), Z = 4.94, P<0.01] and ankle-R2 [WMD = 6.12, 95% CI (4.02, 8.21), Z = 5.72, P<0.01] of children with cerebral palsy. The results showed that there was a significant difference between the two groups, with the experimental group being significantly higher than the control group (S4 and S5 Figs).

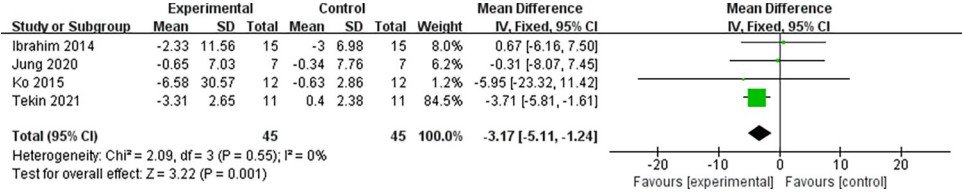

**Fig 7. Effect of WBV training on TUG in children with cerebral palsy.**

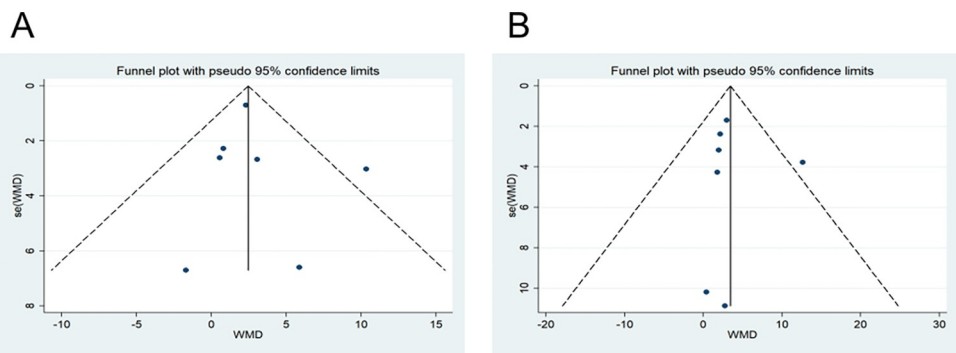

**Fig 8. Funnel diagram of GMFM88-D and GMFM88-E.**

In Begg's test, P = 0.317>0.1, which indicated that there was no publication bias in the literature. In summary, the results showed that WBV training could effectively improve the passive range of motion of the ankle joint and the angle of the ankle joint during muscle reaction in children with cerebral palsy.

## Publication bias assessment

The GMFM88-D and GMFM88-E were used as indicators to plot the inverted funnel diagram, using WMD as the abscissa and SE as the ordinate, as shown in Fig 8. It can be seen from the figures that the graphic distribution of the two funnel charts was basically symmetrical, and the scattered points of each study were also within the scope of the inverted funnel chart. The possibility of literature bias was small, and the analysis results were reliable.

## Discussion

The incidence of cerebral palsy is gradually increasing with an increasingly younger age of onset and approximately 80% disability in survivors [3, 34]. Most patients with cerebral palsy have motor dysfunction due to brain injury or lesions, which affects the lower limb motor function of patients with cerebral palsy [1–3, 34]. Lower limb rehabilitation in children with cerebral palsy is a clinical difficulty. Although there have been a variety of therapies to improve lower limb function, there is a lack of specific means [35]. A single routine rehabilitation therapy process can be monotonous and the content is boring, the lack of interaction between children and therapists can easily lead to a reduction in the children's interest, and the heavy workload can also lead to therapists fatigue [36]. Whole-body vibration training is to promote the recovery of limb and joint functions and enhance muscle strength and limb coordination and flexibility through a warm-up, traction, strength training, muscle relaxation and joint injury rehabilitation [37, 38]. Under certain vibration conditions, WBV training has benefits for some people with cerebral palsy, such as reducing spasticity, improving muscle strength, increasing joint mobility and muscle thickness, promoting postural stability, improving balance ability, and enhancing gross motor ability [39]. Moreover, the greatest feature of WBV training is that it can achieve effective rehabilitation with a small load, without causing excessive burdens on the heart and lung, and with little impact on important organs such as the cardiovascular and nervous system [40]. For CP patients, WBV training is simple, non-invasive, relatively safe, and has a short training time. Even one-time intervention can also produce positive effects, which can better stimulate the interest and enthusiasm for rehabilitation in children. The combination of WBV training and conventional treatment may play a better role in rehabilitation potential, thus achieving better treatment outcomes [41, 42].

Although many scholars have used WBV training to improve the lower limb motor function of children with cerebral palsy and most of them have achieved satisfactory results [43, 44]. However, Nordlund (2007), Ruck (2010) and Pozo Cruz (2012) showed that WBV training is not completely effective in improving muscle and nerve function, and the efficacy of treatment remains to be determined [45–47]. In the literature search, the research group found that many RCTs combined with other therapies could not be meta-analyzed and were therefore excluded. Some of the high-quality papers differed in their evaluation metrics and could not be combined and analyzed on the data. In addition, in previous studies on a systematic review of WBV training in children with cerebral palsy, only electromyographic indicators were mentioned, and no other indicators were mentioned by Pozo Cruz (2012) [47]. Therefore, the research group integrative selected GMFM88-D/E score, 6MWT, TUG, BBS and Ankle function as the indexes to evaluate the lower limb motor function of children with cerebral palsy. The analysis results show that:

1. WBV training effectively improved the gross motor function of lower limbs in children with cerebral palsy.

2. WBV training did not significantly improve the 6MWT walking speed of children with cerebral palsy.

3. WBV training effectively improved the TUG performance of children with cerebral palsy.

4. WBV training effectively improved the balance ability of children with cerebral palsy.

5. WBV training can effectively improve the ankle function of children with cerebral palsy.

In terms of gross motor function, children with cerebral palsy have movement retardation and muscle stiffness in the early stage. Rehabilitation treatment through whole-body vibration training can effectively improve muscle strength and posture control and, in addition, in enhancing the standing ability of patients to walk, run, and jump. This result consistent with previous studies [8], which indicate that the whole-body vibration training has a significant positive impact on the gross motor function of lower limbs.

In terms of 6MWT walking speed, the included literature generally showed no significant difference and great heterogeneity, which means WBV training did not significantly improve the 6MWT walking speed of children with cerebral palsy. However, two of the literatures showed that WBV training was effective and two literatures showed that WBV training was ineffective, which was found by comparison, indicating that the experimental sample size and treatment period of literatures that WBV training was effective were larger than literatures that showed that WBV training was ineffective. Through the analysis of the included original articles, we found that the two articles did not combine conventional training and WBV as treatment methods, and different treatment plans may have different results. In Ahmadizadeh (2019), after the intervention of WBV training, the 6MWT performance of the experimental group was not higher than the control group. The reason may be that, although the results showed that WBV training could improve the muscle strength and coordination, balance and walking speed of children with CP, However, WBV training did not significantly increase the range of motion of knee extension and the passive range of motion of hip flexion, abduction and ankle extension. The experiment also showed that the combination of stretching and WBV did not change the severity of knee muscle spasm in CP children, which may have resulted in no statistically significant difference in 6MWT performance between the experimental group and the control group. Additionally, in Jung (2020), the 6MWT performance increased between the two groups, but the change was not obvious. The reason may be that the control group used WBV training, and the experimental group added action observation.

Participants were instructed to watch a video on a 17-inch laptop screen from a distance of 50 cm. During this time, they were instructed to either follow or to not follow the actions demonstrated in the video. However, due to the problems of the experimental group, such as the short attention span of action observation and the unqualified action requirements, the changes of the two groups may not be obvious. Moreover, short intervention time, small study sample size, limited application frequency of WBV, weak control of other factors and other problems, which may also lead to WBV training did not significantly improve 6MWT performance. Cheng (2015) also explained that the improvement in 6MWT was significant, but it gradually disappeared after the intervention stopped [48]. Thus, long-term intervention may be required to achieve more sustained improvements. The differences in the experimental sample size, number of cycles, choice of treatment regimen as well as the measures in RCTs may affect the overall stability, therefore, more experimental studies are needed to verify the effectiveness of WBV training on 6MWT walking speed in children with cerebral palsy.

In terms of TUG performance, the indicators included in the literature indicate that WBV training can effectively improve the TUG performance of children with cerebral palsy, possibly because WBV training enhances the coordination and flexibility of lower limbs, reduces the risk of falls and shortens the testing time of children with cerebral palsy, which is consistent with previous studies [49].

In terms of balance ability, the included literatures differed significantly with large heterogeneity, and after sensitivity analysis, the heterogeneity decreased. In addition, the results of the two groups were also statistically different. Therefore, WBV training can effectively improve the balance ability of children with cerebral palsy. Balance training with different vibration modes and methods, multiple functional activities require patients to actively transfer their center of gravity to maintain dynamic or static balance, which is consistent with previous studies [39, 50], indicating that WBV training can improve patients' lower limb weight-bearing, balance and walking ability.

In terms of ankle function, WBV training could significantly improve the passive range of motion and ankle angle of the ankle joint during muscle reaction in children with cerebral palsy, but there was no significant difference in the active range of motion of the ankle joint, the reason may be that the differences in the age, sample size, and treatment period of the subjects selected in the two articles resulted in variations in the results Therefore, more experimental studies are needed to verify the effectiveness of WBV training on the active range of motion of the ankle in children with cerebral palsy. On the whole, whole-body vibration training still improved the ankle function of children with cerebral palsy, which is consistent with previous studies [41].

## Limitation

This study only includes the published randomized controlled studies and does not include the review and conference literature, which may have a certain impact on the results. Future studies can add inclusion criteria on this basis. Moreover, the research quality of the literature included in this paper is not high. It is suggested that the clinical trial design should be stricter to improve the quality of the research.

The overall sample size of this study is small. The difference between the intervention group and the control group and the intervention cycle may be one of the main reasons for the statistical deviation. It is generally believed that the longer the experimental cycle, the more obvious the therapeutic effect. There are 8 literature whose experimental period is less than 10 weeks, which may affect the overall combined analysis. Therefore, it is suggested that the duration of intervention measures in future research should be 10 weeks as a reference.

At present, the clinical trials of WBV training on the rehabilitation of lower limb motor function are complex and diverse, so that the existing clinical evidence is difficult to meet the actual needs, and the accuracy of the conclusion is significantly reduced. There are few included studies on BBS and Ankle function. In the future, relevant research can appropriately increase and refine the discussion and research of these two indicators. In addition, there is no unified standard for the optimal value of vibration mode, amplitude and frequency in WBV training, which is also a direction for future research.

## Conclusion

In conclusion, the analysis results show that WBV training combined with conventional physiotherapy can improve the lower limb motor function of children with cerebral palsy more than conventional physiotherapy alone. This study followed the results of evidence-based medical research and used meta-analysis to combine the results of clinical RCTs on the effects of WBV training on lower limb motor function in children with cerebral palsy. This article overcomes the shortcomings of small experimental samples and incomplete consistency of results and improves the statistical test efficacy. It provides more reliable evidence for clinical practice and decision making of WBV training rehabilitation for children with cerebral palsy.

## Supporting information

**S1 Checklist. PRISMA 2009 checklist.**
(DOC)

**S1 Table. Search strategy.**
(DOCX)

**S1 Fig. Effect of WBV training on BBS balance ability of children with cerebral palsy.**
(TIF)

**S2 Fig. Effect of WBV training on Ankle-active ROM in children with cerebral palsy.**
(TIF)

**S3 Fig. Effect of WBV training on Ankle-passive ROM in children with cerebral palsy.**
(TIF)

**S4 Fig. Effect of WBV training on Ankle-R1 in children with cerebral palsy.**
(TIF)

**S5 Fig. Effect of WBV training on Ankle-R2 in children with cerebral palsy.**
(TIF)

## Acknowledgments

This work was supported by the Gdansk University of Physical Education and Sport. The authors thank Dr. Cheng for her in-depth discussion on the effect of WBV training on lower limb motor function in children with cerebral palsy.

## Author Contributions

**Conceptualization:** Xiaoye Cai.

**Data curation:** Xiaoye Cai.

**Funding acquisition:** Zbigniew Ossowski.

**Investigation:** Guoping Qian.

**Methodology:** Xiaoye Cai, Guoping Qian.

**Project administration:** Zbigniew Ossowski.

**Resources:** Guoping Qian, Feng Wang.

**Software:** Xiaoye Cai.

**Supervision:** Feng Wang.

**Validation:** Siyuan Cai.

**Visualization:** Yingjuan Da.

**Writing – original draft:** Xiaoye Cai, Guoping Qian.

**Writing – review & editing:** Siyuan Cai, Zbigniew Ossowski.

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
