## [Decision Letter · Decision Letter 0]

21 Nov 2022

PONE-D-22-11724The effect of whole-body vibration on lower extremity function in children with cerebral palsy: a meta-analysisPLOS ONE

Dear Dr. Ossowski,

Thank you for submitting your manuscript to PLOS ONE. After careful consideration, we feel that it has merit but does not fully meet PLOS ONE’s publication criteria as it currently stands. Therefore, we invite you to submit a revised version of the manuscript that addresses the points raised during the review process.

We look forward to receiving your revised manuscript.

Kind regards,

Nili Steinberg

Academic Editor

PLOS ONE

Journal Requirements:

Additional Editor Comments (if provided):

Please see reviewers' comments

Reviewers' comments:

Reviewer's Responses to Questions

**Comments to the Author**

1. Is the manuscript technically sound, and do the data support the conclusions?

Reviewer #1: Yes

Reviewer #2: Yes

2. Has the statistical analysis been performed appropriately and rigorously? 

Reviewer #1: Yes

Reviewer #2: No

3. Have the authors made all data underlying the findings in their manuscript fully available?

Reviewer #1: Yes

Reviewer #2: Yes

4. Is the manuscript presented in an intelligible fashion and written in standard English?

Reviewer #1: Yes

Reviewer #2: No

5. Review Comments to the Author

Reviewer #1: Cai et al. present a well written and thoughtful meta analysis on the use of WBV for lower extremity function in patients with cerebral palsy. I commend the authors for the thoroughness of their analysis and presentation of their findings in a reasonable fashion. I think the discussion did cover the salient points of further consideration based on the authors' findings. I also found the conclusion to be reasonable.

Reviewer #2: The manuscript is a meta analysis study that investigates the effect of whole-body vibration (WBV) as an intervention approach on motor function in the lower limb among children with cerebral palsy (CP). For the purposes of standardization the study only reviewed articles that incorporated a randomised control trials to evaluate the effect of WBV training on rehabilitation of lower limb function. The manuscript is suitable of the Plos One, is timely, and investigates an important and relevant issue of lower limb rehabilitation in children with CP. My main concerns are poor description of the search strategy, lack of clarity on the statistical analysis and the lack of interpretation and in-depth discussion of the findings on why whole-body vibration would improve lower limb function and performance.

General comments:

#1: Please refer to the supplementary material here!! I had to go back and search for it specifically as the description provided in text under 2.1 section is simply incomplete and unclear.

The current description under section 2.1 needs to be adapted.

In addition, within the supplementary material please justify the use of search terms that are marred with redundancies e.g. “Cerebral Palsy” used as a search term individually, but also in combination with other subtypes. This strategy is not very robust. Furthermore, the use of “… Palsy” vs “… Palsies” (search term #5) isn’t recommended, instead choosing truncation with * might be a preferred approach. Did you check what happens to searches when using truncation? Please specify in the supplementary material document. Also you might want to consider presenting the information in the Supplementary material as a table to ensure clarity.

#2: Section 2.5 on Statistical analysis is poorly described. Use of phrases such as “…trial…” in the first paragraph, vs. “In this case, the fixed model was adopted”. What trials, and what was the model? Was generalised linear model considered? If so what was/were the dependent (vs independent) variable/s in the model? What was the mean difference weighted with? Sample size? Please specify.

#3. In the results section 3.3. What is Z? The overall effect size? Please specify. Please also consider denoting Z in the figures on the horizontal axis, this will allow better interpretability of findings. Also in the first paragraph what is I2? Or is this simply a typo and represents heterogeneity? In methods P is specified in italics and in results simply as P. Are they the same things? Please carefully read the text to ensure standardisation.

#4: When authors talk about studies with WBV intervention, this is in fact combined with conventional training in almost all cases. Are there reports where WBV has been used alone as therapy (probably not so many given that physiotherapy might be standard of care in CP)? Does the meta-analytic approach presented here account for the additive effects? Please specify. This has larger implications for undertaking such a therapy in the future. This approach is then complementary to the traditional conventional physiotherapy - its efficacy therefore must always be interpreted with care. Please consider including this aspect in discussion. Finally, sentences (Discussion 1st paragraph) such as “conventional therapy is time-consuming, single modality…” are difficult to interpret, given that the systematic review reports on studies that have undertaken WBV + conventional training, right? Wouldn’t they be even more time consuming?

#5: In addition to #1: recommendations/guidelines on use of WBV training in CP? It would nicely complement the current text.

#6: The gross motor function measurement (GMFM)-88-D and E scores improved, but 6 minute walk test performance didnt. Is an interesting finding, please elaborate on why authors think this would be the case?

Specific comments:

S#1: Too many illustrations makes it difficult to focus on the take home message. Certain parameters e.g. Berg balance scale and ankle function, have only been undertaken in 3 or 4 studies and are secondary outcomes. These can either be combined, or simply removed (and included as supplementary material) with the relevant findings only presented as text. Figures 13 and 14 can also be combined with the use of two different types of markers/points. These changes will enhance the readability of the text and interpretability of the outcomes.

S#2: While the manuscript is generally comprehensible, there is still a lot of room for improvement. Specific cases would be e.g.

a. Please rephrase“home and abroad…” (Introduction, 2nd paragraph).

b. Section 2.5 Statistical Analysis 2nd paragraph “Considering the factors that might lead to heterogeneity, there also might be heterogeneity” what does that mean?

c. Statistical Analysis 3rd paragraph “For studies differing considerably from other included studies in methodology or findings, a sensitivity was conducted, and those studies were excluded from the meta-analysis. What is considerably? Please clearly specify.

And many other such phrases/sentences make it difficult to comprehend, what the authors are referring to. Consider revising the text for enhanced interpretation.

6. PLOS authors have the option to publish the peer review history of their article (what does this mean?). If published, this will include your full peer review and any attached files.

Reviewer #1: **Yes: **Albert Tu

Reviewer #2: **Yes: **Navrag B. Singh

---

## [Author Response · Author response to Decision Letter 0]

2 Jan 2023

Reply to Reviewer #2

Dear Reviewer,we very much appreciate the time and effort you have put into your comments. Your advice about the formatting of my paper is most helpful.

Comments:

“The manuscript is a meta analysis study that investigates the effect of whole-body vibration (WBV) as an intervention approach on motor function in the lower limb among children with cerebral palsy (CP). For the purposes of standardization the study only reviewed articles that incorporated a randomised control trials to evaluate the effect of WBV training on rehabilitation of lower limb function. The manuscript is suitable of the Plos One, is timely, and investigates an important and relevant issue of lower limb rehabilitation in children with CP. My main concerns are poor description of the search strategy, lack of clarity on the statistical analysis and the lack of interpretation and in-depth discussion of the findings on why whole-body vibration would improve lower limb function and performance.”

We appreciate your clear and detailed feedback and hope that the explanation has fully addressed all your concerns. In the remainder of this letter, we discuss each of your comments individually along with our corresponding responses.

To facilitate this discussion, we first retype your comments in italic font and then present our response to the comments.

Comment 1:

Please refer to the supplementary material here !! I had to go back and search for it specifically as the description provided in text under 2.1 section is simply incomplete and unclear.

Response 1:

Many thanks for your comment. Our previous expression of the search strategy was not very clear and we have revised the search strategy in detail according to your comment.

Comment 2:

The current description under section 2.1 needs to be adapted.

Response 2:

Many thanks pointing out this problem. We have updated the expression of 2.1 to make the article more readable and easier for readers to understand.

Comment 3:

In addition, within the supplementary material please justify the use of search terms that are marred with redundancies e.g. “Cerebral Palsy” used as a search term individual, but also in combination with other subtypes. This strategy is not very robust.Furthermore, the use of”...Palsy”vs”...Palsies”(search term #5) isn’t recommended,instead choosing truncation with * might be a preferred approach. Did you check what happens to searches when using truncation? Please specify in the Supplementary material document. Also you might want to consider presenting the information in the Supplementary material as a table to ensure clarity.

Response 3:

Many thanks for your comment. We re-conducted literature search in major databases to check whether any new articles meeting the criteria were included in this study. Based on your comment, A renewed literature search was carried out based on the use of truncation combined with subject terms and free words, and in the end, the number of included articles in the same number as originally, and the search strategy is presented in the table in the supplementary material. In addition, we have updated the deadline for performing database searches.

Comment 4:

Section 2.5 on Statistical analysis is poorly descried.Use of phrases such as ”...trial...”in the first paragraph,vs ,”In this case,the fixed model was adopted”.What trials, and what was the model? Was generalised linear model considered? If so what were the dependent (vs independent)variable/s in the model? What was the mean difference weighted with? Sample size? Please specify.

Response 4:

Many thanks for the advice you have provided. Our previous expression of statistical analysis methods was not very clear enough and we have revised them in detail according to your suggestion. For example, For continuous variables, the weighted mean difference (WMD) was used as the effect index, and each effect size was expressed with a 95% confidence interval (95%CI), or When I2 ≤ 50%, this indicates good homogeneity across studies and the effect sizes were combined using a fixed effects model. Alternatively, for times when heterogeneity was large (I2 > 50%) we also considered constructing a linear regression model, but heterogeneity was not reduced by regression analysis, so only sensitivity analyses were conducted when discussing sources of article heterogeneity.

Comment 5:

In the results section 3.3. What is Z? The overall effect size? Please specify.Please also consider denoting Z in the figures on the horizontal axis, this will allow better interpretability of findings. Also in the first paragraph what is I2? Or is this simply a typo and represents heterogeneity? In methods P is specified in italics and in results simply as P. Are they the same things? Please carefully read the text to ensure standardisation.

Response 5:

Many thanks for pointing out this issue. We have carefully checked the entire manuscript and corrected all spelling errors to ensure standardisation. Also according to your suggestion, we have changed the software (Review Manager5.3) and updated all the images. The overall effect “Z” is shown in the images.

Comment 6:

When authors talk about studies with WBV intervention, this is in fact combined with conventional training in almost all cases. Are there reports where WBV has been used alone as therapy (probably not so many given that physiotherapy might be standard of care in CP)? Dose the meta-analytic approach presented here account for the additive effects? Please specify. This has larger implications for undertaking such a therapy in the future. This approach is then complementary to the traditional conventional physiotherapy- its efficacy therefore must always be interpreted with care. Please consider including this aspect in discussion. Finally,sentences (Discussion 1st paragraph) such as ”conventional therapy is time-consuming,single modality...”are difficult to interpret, given that the systematic review reports on studies that have undertaken WBV+conventional training, right? Wouldn’t they be even more time consuming?

Response 6:

Many thanks for your comment. According to your comment, we have rechecked the details of the included articles and additionally searched whether there were any new reports of WBV being used as a treatment Separately.We did not find new articles that met this criterion, and the articles that were included were indeed a combination of conventional treatment + WBV training. In the case of WBV training as a complementary therapy, we have searched related literature and interpreted its efficacy with caution, taking into account its additive effects, and have included the findings in the discussion section. In addition, the question of why WBV therapy is less time-consuming than conventional therapy is explained as follows the monotonous and boring of single conventional therapy, the lack of interaction between the child and the therapist, and the tendency for the child to become less interested and less cooperative, which can lead to longer sessions. The addition of WBV training as an adjunct to therapy is a short, one-off intervention that has a positive effect, stimulates the child's interest and motivation, and the combination of the two can also give full play to the potential of rehabilitation, so as to achieve better therapeutic effect.

Comment 7:

In addition to # 1: recommendations/ guidelines on use of WBV training in CP? It would nicely complement the current text.

Response 7:

Thank you for this valuable comment. We have searched and read the relevant studies and found that there are no clear guidelines on the use of WBV training for CP, but we have clearly presented some relevant recommendations or research results on the use of WBV training for CP in our manuscript.

Comment 8:

The gross motor function measurement (GMFM)-88-D and E scores improved,but 6 minutes walk test performance didnt. Is an interesting finding,please elaborate on why authors think this would be the case?

Response 8:

Many thanks for your comment. According to your suggestion, we have detailed the fact that there was no significant difference in 6MWT performance between the control and experiment groups in the discussion section.

Comment 9:

Too many illustrations makes it difficult to focus on the take home message. Certain parameters e.g. Berg balance scale and ankle function, have only been undertaken in 3or4 studies and are secondary outcomes.These can either be combined, or simply removed (and included as supplementary material) with the relevant findings only presented as text.Figures 13 and 14 can also be combined with the use of two different types of markers/points. These changes will enhance the readability of the text and interpretability of the outcomes.

Response 9:

Thank you for pointing this out, we have included images of the secondary outcomes (BBS and ankle function) in the supplementary material and the relevant findings are presented in text only. The two funnel plots are combined and represented as A and B to enhance the readability of the text and the interpretability of the results.

Comment 10:

While the manuscript is generally comprehensible there is still a lot of room for improvement. Specific cases would be e.g. 

a.Please rephrase"home and abroad..."(Introduction, 2nd paragraph).

Response 10:

Many thanks for pointing out this issue. According to your suggestion, we have rephrased the second paragraph of the introduction.

Comment 11:

b.Section 2.5 Statistical Analysis 2nd paragraph"Considering the factors that might lead to heterogeneity, there also might be heterogeneity" what does that mean?

Response 11:

Thank you for pointing this out. The implication of this statement is that there is heterogeneity between studies and that sources of heterogeneity need to be sought. We considered sensitivity analysis and subgroup analysis, meta-regression.

Comment 12:

c.Statistical Analysis 3rd paragraph "For studies differing considerably from other included studies in methodology or findings, a sensitivity was conducted, and those studies were excluded from the meta-analysis. What is considerably? Please clearly specify.

Response 12:

Thank you for raising this issue. The implication of this statement is that there is heterogeneity between studies and sources of heterogeneity need to be sought. We considered sensitivity analysis and subgroup analysis, meta-regression. For example, When P≤0.1 and I2 > 50%，it means that there was heterogeneity between the studies.The source of heterogeneity was identified, and sensitivity analysis was conducted by article by article elimination. If I2≤50% after deleting a single study,the study was considered to be more different from other studies and might be the source of influencing the combined effect size, and it was excluded from the meta-analysis.

Comment 13:

And many other such phrases/sentences make it difficult to comprehend, what the authors are referring to. Consider revising the text for enhanced interpretation.

Response 13:

Thank you very much for your comment. We have read it carefully and revised the wording and sentences to improve the whole text.

We would like to take this opportunity to thank you for all your time involved and this great opportunity for us to improve the manuscript. We hope you will find this revised version satisfactory.

Sincerely,

The Authors

Reply to Reviewer #1

Dear Reviewer,

Thank you very much for your time involved in reviewing the manuscript and your very encouraging comments on the merits.

Comments:

“Cai et al. present a well written and thoughtful meta analysis on the use of WBV for lower extremity function in patients with cerebral palsy. I commend the authors for the thoroughness of their analysis and presentation of their findings in a reasonable fashion. I think the discussion did cover the salient points of further consideration based on the authors' findings. I also found the conclusion to be reasonable.”

We also appreciate your clear and detailed feedback and hope that the explanation has fully addressed all your concerns. In the remainder of this letter, we discuss each of your comments individually along with our corresponding responses.

We hope that the changes we have made resolve all your concerns about the article. We are more than happy to make any further changes that will improve the article and/or facilitate successful publication.

Sincerely, 

The Authors

---

## [Editor Report · Decision Letter 1]

20 Feb 2023

The effect of whole-body vibration on lower extremity function in children with cerebral palsy: a meta-analysis

PONE-D-22-11724R1

Dear Dr. Ossowski,

We’re pleased to inform you that your manuscript has been judged scientifically suitable for publication and will be formally accepted for publication once it meets all outstanding technical requirements.

Kind regards,

Nili Steinberg

Academic Editor

PLOS ONE
---

## [Editor Report · Acceptance letter]

2 Mar 2023

PONE-D-22-11724R1 

The effect of whole-body vibration on lower extremity function in children with cerebral palsy: a meta-analysis 

Dear Dr. Ossowski:

I'm pleased to inform you that your manuscript has been deemed suitable for publication in PLOS ONE. Congratulations! Your manuscript is now with our production department. 

Kind regards, 

on behalf of

Prof. Nili Steinberg 

Academic Editor

PLOS ONE